# *Nephelium lappaceum* Extract as an Organic Inhibitor to Control the Corrosion of Carbon Steel Weldment in the Acidic Environment

Femiana Gapsari [1,*], Djarot B. Darmadi [1], Putu H. Setyarini [1], Hubby Izzuddin [2], Kartika A. Madurani [3], Ayoub Tanji [4] and Hendra Hermawan [4]

[1] Department of Mechanical Engineering, Faculty of Engineering, Brawijaya University, MT Haryono 167, Malang 65145, Indonesia; b_darmadi_djarot@ub.ac.id (D.B.D.); putu_hadi@ub.ac.id (P.H.S.)
[2] Research Center for Physics, National Research and Innovation Agency, Bld 440-442 Komplek Puspiptek Serpong, Tangerang Selatan 15314, Indonesia; hubhubb001@lipi.go.id
[3] Department of Chemistry, Faculty of Science and Data Analytics, Institut Teknologi Sepuluh Nopember, Arief Rahman Hakim, Surabaya 60111, Indonesia; kartika.an@gmail.com
[4] Department of Mining, Metallurgical and Materials Engineering, Laval University, Quebec City, QC G1V 0A6, Canada; ayoub.tanji.1@ulaval.ca (A.T.); hendra.hermawan@gmn.ulaval.ca (H.H.)
[*] Correspondence: memi_kencrut@ub.ac.id; Tel.: +62-822-3644-1750

**Abstract:** Organic inhibitors have been considered as an effective way to control the corrosion of carbon steel weldment in an acidic environment. This work proposes a new green organic inhibitor made of extract of rambutan fruit (*Nephelium lappaceum*) peel and aims at analyzing its corrosion inhibitor properties and protection mechanism. Specimens of carbon steel weldment were tested for their corrosion by using electrochemical and immersion methods in 1 M HCl solution containing 0 to 6 g/L of Nephelium peel (NP) extract. Results showed that, in the same solution, the corrosion rate was measured to be higher on the weld metal zone than that of base metal zone, which could be related to the coarser grain of the weld metal zone and the stability of the formed oxide layer. The addition of NP extract was found to increase the stability of the oxide layer, thus increasing the corrosion resistance of the specimens. The maximum inhibition efficiency of the NP extract was reached at 97% for weld metal with 5 g/L of extract, at 80% for the heat affected zone with 5 g/L, and at 70% for base metal with 4 g/L. This work reveals the particularity of different weldment zones to the different needs of inhibitor concentration for obtaining the optimum corrosion protection.

**Keywords:** acidic environment; corrosion; electrochemical test; *Nephelium lappaceum*; organic inhibitor; weldment

## 1. Introduction

In a chloride-containing acidic environment, corrosion becomes the principal cause of many failures in steel structures [1,2]. The aggressive chloride ions ($Cl^-$) provoke localized corrosion attacks at any site having microstructural heterogeneity, including steel weldment. Compositional segregation, residual stress, and microstructural variation in the welding zone lower the corrosion resistance of the weldment [3]. Microstructural variation, such as grain size and shape, influences the active passive response of the corroded surface of a weldment, where a finer grain may lead to a lower rate of localized corrosion but a higher rate of uniform corrosion [4]. Weldment corrosion in a chloride-containing acidic environment can be controlled by employing inhibitors. This inhibitor should be able to reduce the susceptibility of hydrogen uptake that is released during the cathodic reaction, thus reducing the risk of hydrogen-induced cracking in the weldment [5].

An effective inhibitor will form an adsorption layer on the metal surface that prevents the aggressive species from attacking the weldment [6]. However, every inhibitor has different suitability to different metals, including different zones of weldment [7], and to different corrosive environments [8]. In a chloride-containing acidic environment, studies showed that many inhibitors exhibit low efficiency in reducing the corrosion of steel [9–14].

Most of them have low biodegradability and are potentially toxic to the environment [11]. Natural organic inhibitors have been viewed as more sustainable and environmentally friendly alternatives to synthetic organic inhibitors. Chemically, they contain S, N, O, P, and ring structures of heterocyclic compounds [10], similar compounds to those that can be found in the rambutan fruit skin or peel (*Nephelium lappaceum*). Rambutan is a typical Indonesian plant and its abundant peel and seed waste are commonly used as bioactive compounds in food, pharmaceutical, and cosmetic industries [15].

Extract of *Nephelium lappaceum* is rich in antioxidant flavonoid compounds [16],ande its multiple polar atoms and electron-rich bonds can promote antioxidant molecule adsorption on metal surfaces by donating electrons to iron atoms and forming coordinate bonds. This extract should have a potential for becoming an effective inhibitor toward corrosion of steel in an aggressive environment. Therefore, this work aims to synthesize a new inhibitor based on *Nephelium lappaceum* extract and to evaluate its effectiveness for protecting carbon steel weldment against corrosion in an aggressive chloride-containing acidic environment.

## 2. Materials and Methods

### 2.1. NP Extract Preparation

A batch of 1500 g of *Nephelium lappaceum* peel (NP) was oven-dried at a temperature of 40–45 °C and then ground into powders. The powders were macerated in ethanol for 72 h and then filtered out to obtain the NP extract. The chemical compound and the extract substance were identified using a Fourier-Transform Infrared Spectroscope (FTIR, Spectrum Two, Perkin Elmer, Buckinghamshire, UK) and a Gas Chromatography Mass Spectrometer (GCMS, Perkin Elmer, UK).

### 2.2. Weldment Preparation

Samples of weldment were prepared by employing a Shielded Metal Arc Welding (SMAW) technique on AISI 1040 carbon steel plates (0.38 wt% C, 0.6 wt% Mn) E6013 type electrodes. The AWS A5.1 standard was followed, where 24 V of arc voltage and 90 A of welding current were applied on the butt weldment with double V single pass on each side. Three zones of interest, namely base metal (BM), heat-affected zone (HAZ), and weld metal (WM) were identified, and both of their macro- and micro-structures were photographed using metallurgical microscopes (Olympus model DSX510 and BX51M, Tokyo, Japan). Then, the microstructure was analyzed using the Jefriss grain count method following the ASTM E112 standard; the hardness of each zone was measured by using a Vickers hardness tester (Future Tech, FLV AR0045, Kanagawa, Japan).

### 2.3. Corrosion Testing

Specimens for corrosion testing were cut from the weldment samples at the three zones of interest and ground with abrasive papers up to 2000 grit. A range of corrosion testing solutions was prepared by dissolving up to 6 g/L (0, 1, 2, 3, 4, 5 and 6) of the NP extract into 1M HCl. Two electrochemical test methods were performed: electrochemical impedance spectroscopy (EIS) and potentiodynamic polarization (PP), using an Autolab potentiostat (PGSTAT 128N, Herisau, Switzerland). A three-electrode setting was used with the weldment specimen as the working electrode, Ag/AgCl (3M KCl) as the reference electrode, and platinum as the counter electrode. Prior to the test, the system was stabilized for 30 min until the open circuit potential (OCP) was recorded. The EIS was conducted at the OCP in the frequency range from 1000 Hz to 0.1 Hz using a signal amplitude of 1 mA peak to peak galvanostatiscally, while the applied potential of the PP test was measured at ±100 mV from $E_{corr}$ at a scan rate of 1 mV/s. The Tafel plot extrapolation and EIS spectra were analyzed by using NOVA 1.11 software associated with the potentiostat.

The corrosion rate (*CR*), in mm/year, was calculated based on Equation (1) [17]:

$$CR = 8.76 \times \frac{W}{\rho A t} \tag{1}$$

where W is weight loss (g), t is exposure time (hour), A is the exposed surface area (which is 1 cm$^2$), and ρ is the density of metal (which is 7.85 g/cm$^3$). The inhibition efficiency (IE) was calculated by using Equations (2) and (3) [18,19].

$$\text{IE } (\%) = \frac{i_{corr} - i_{corr}{}^0}{i_{corr}} \times 100\% \tag{2}$$

$$\text{IE } (\%) = \frac{R_{ct} - R_{ct}{}^0}{R_{ct}} \times 100\% \tag{3}$$

where $i_{corr}$ and $i_{corr}{}^0$ are the corrosion current density obtained from the PP test with and without the addition of NP extract; $R_{ct}$ and $R_{ct}{}^0$ are the charge transfer resistance obtained from EIS analysis with and without the addition of NP extract. To evaluate the NP extract inhibition mechanism, a longer corrosion test was carried out by statically immersing the specimen for 10 days in two testing solutions, 1M HCl and 1M HCl with the addition of 4 g/L and 5 g/L of NP extract.

## 3. Results

### 3.1. Weldment Structure and Hardness

Figure 1 reveals a typical macro and microstructure of the samples after welding. The three zones of interest, weld metal (WM), heat-affected zone (HAZ), and base metal (BM) are shown in Figure 1a. The microstructure of WM (Figure 1b) and HAZ (Figure 1c) are characterized by perlite and ferrite phases in various forms: acicular, polygonal, grain boundaries, and Widmanstatten structures. The grain boundary ferrite dominates the WM zone while acicular ferrite dominates the HAZ zone. Some precipitates are also observed in the WM microstructure. The BM zone (Figure 1d) consists of perlite and ferrite phases with an average grain size of 10.5 μm, finer than that of HAZ of 12.5 μm and WM of 21.7 μm. The BM zone possesses the highest average hardness at 248.9 VHN, followed by the HAZ at 207.6 VHN, and the WM zone at 164.2 VHN.

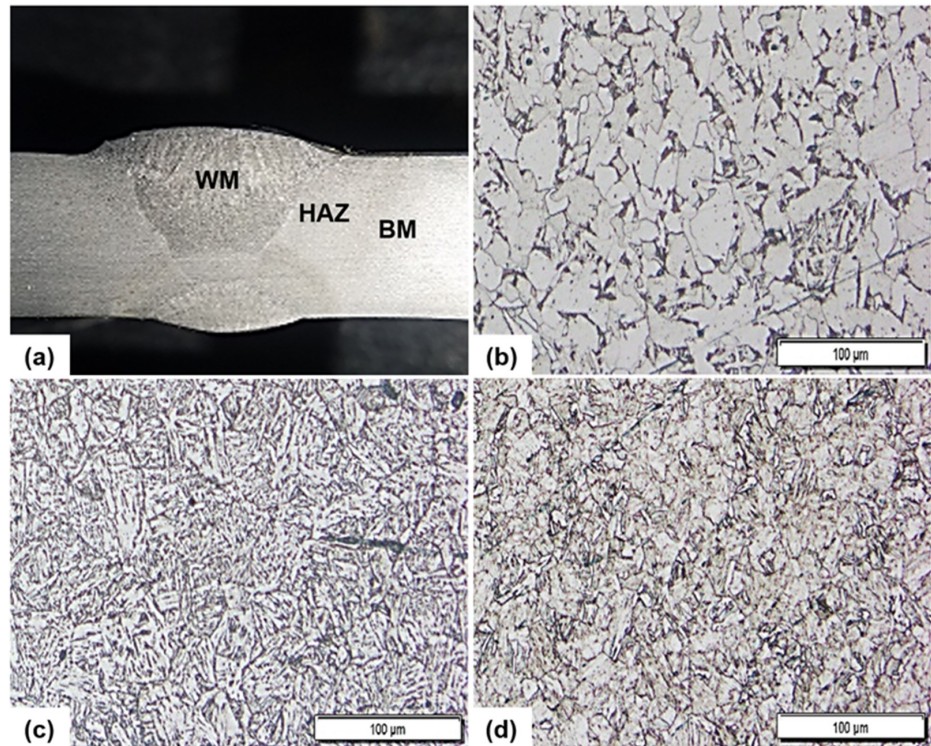

**Figure 1.** (**a**) Macrograph of welded samples, (**b**) weld metal (WM), (**c**) heat-affected zone (HAZ), (**d**) base metal (BM).

### 3.2. NP Extract Characteristics

Figure 2 shows an FTIR spectrum of the NP extract which indicates the appearance of several distinctive functional organic groups, which are O-H bond (3411 cm$^{-1}$), C=O bond (2076 cm$^{-1}$), C=C double bond (1707 and 1619 cm$^{-1}$), and C-H bond (1515 and 1446 cm$^{-1}$). In addition, C-O, =C-O, =C-H, and aromatic C-H bonds are also present at 1345, 1216, 1050, 973, 865, and 604 cm$^{-1}$, respectively. Further identification of the organic compounds was done by using the GCMS technique as shown in Table 1. This reveals that the NP extract mostly consisted of some inositol component (63.20%) which showed up at retention time of 11.015–11.570 min with molecular weight of 180 g/mol. Other components are also identified in the extract as heptadecene-(8) carbonic acid of 17.43% at retention time 12.660 min with a molecular weight of 268.4 g/mol; 1,2-benzenediol of 1.83% at retention time 9.003 min with a molecular weight of 110 g/mol; and 1,2,3-benzenetriol of 1.83% at retention time 9.8880 min with a molecular weight of 126 g/mol. All these components correspond to molecular structures with–OH, C=O, C=C, C-H, =C-H, and aromatic bonds as illustrated by the FTIR adsorption bands.

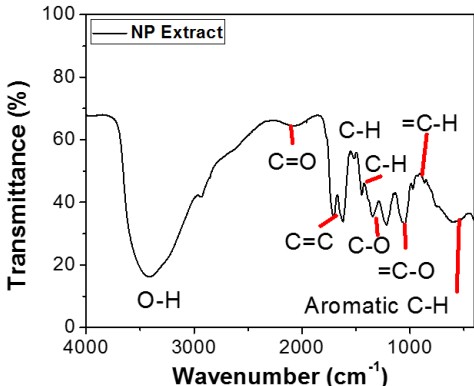

**Figure 2.** FTIR spectra of NP extract.

**Table 1.** Organic compound identification of NP extract by using GCMS.

| Peak | R Time | Area% | Name | Structural Formula | Molecular Weight |
|------|--------|-------|------|--------------------|------------------|
| 1 | 9.003 | 3.12 | 1,2-Benzenediol | | 110 |
| 2 | 9.280 | 2.22 | Unidentified | | |
| 3 | 9.880 | 1.83 | 1,2,3-Benzenetriol | | 126 |
| 4 | 10.377 | 3.85 | Unidentified | | |
| 5 | 11.015 | 2.44 | Mome Inositol | | |
| 6 | 11.263 | 25.02 | Mome Inositol | | 180 |
| 7 | 11.570 | 35.74 | Mome Inositol | | |
| 8 | 11.784 | 3.97 | Unidentified | | |
| 9 | 12.373 | 4.38 | Unidentified | | |
| 10 | 12.660 | 17.43 | Heptadecene-(8)-carbonic acid-(1) | | 268.4 |

### 3.3. Potentiodynamic Polarization

Figure 3 depicts typical potentiodynamic polarization curves of the specimens tested under different concentrations of NP extract. For the BM zone (Figure 3a), the curves shift

toward more negative potentials and lower current densities which are not that obvious for those of the HAZ (Figure 3b) and the BM zone (Figure 3c). Table 2 details the corrosion parameters that were calculated based on the Tafel extrapolation. The NP extract appeared to give a strong inhibition effect to the WM zone where the current density decreased as the extract concentration increased. For this zone, the lowest corrosion rate of 0.67 mm/year was reached at the extract concentration of 5 g/L, giving an inhibition efficiency of 97%. The same extract concentration gave the highest efficiency of 80% for HAZ with a corrosion rate of 0.78 mm/year. For the BM zone, the highest inhibition efficiency was reached at 70% for the extract concentration of 4 g/L, giving a corrosion rate of 1.23 mm/year.

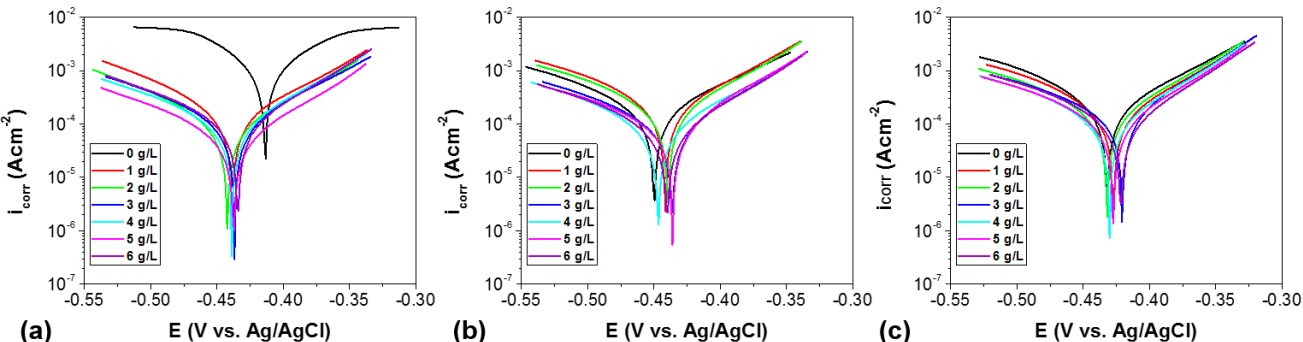

**Figure 3.** Typical potentiodynamic polarization (PP) curves of: (**a**) WM, (**b**) HAZ, (**c**) BM for different concentrations of NP extract.

**Table 2.** Corrosion parameters calculated by using Tafel extrapolation.

| NP Extract (g/L) | $\beta_a$ (V/dec) | $-\beta_c$ (V/dec) | $E_{corr}$ (V) | $i_{corr}$ (A/cm$^2$) | Corrosion Rate (mm/year) | IE (%) |
|---|---|---|---|---|---|---|
| WM | | | | | | |
| 0 | 0.096 | 0.111 | −0.414 | $1.64 \times 10^{-3}$ | 19.3 | 0 |
| 1 | 0.101 | 0.105 | −0.438 | $1.81 \times 10^{-4}$ | 2.13 | 89 |
| 2 | 0.081 | 0.081 | −0.442 | $9.10 \times 10^{-5}$ | 1.07 | 94 |
| 3 | 0.075 | 0.066 | −0.437 | $8.06 \times 10^{-5}$ | 0.95 | 95 |
| 4 | 0.068 | 0.052 | −0.439 | $6.52 \times 10^{-5}$ | 0.77 | 96 |
| 5 | 0.094 | 0.079 | −0.438 | $5.72 \times 10^{-5}$ | 0.67 | 97 |
| 6 | 0.137 | 0.101 | −0.434 | $1.41 \times 10^{-4}$ | 1.66 | 91 |
| HAZ | | | | | | |
| 0 | 0.204 | 0.143 | −0.450 | $3.41 \times 10^{-4}$ | 4.01 | 0 |
| 1 | 0.105 | 0.102 | −0.441 | $2.49 \times 10^{-4}$ | 2.93 | 27 |
| 2 | 0.117 | 0.085 | −0.440 | $2.04 \times 10^{-4}$ | 2.40 | 40 |
| 3 | 0.096 | 0.071 | −0.436 | $8.23 \times 10^{-4}$ | 0.97 | 76 |
| 4 | 0.084 | 0.076 | −0.447 | $7.54 \times 10^{-4}$ | 0.89 | 78 |
| 5 | 0.086 | 0.058 | −0.436 | $6.69 \times 10^{-4}$ | 0.78 | 80 |
| 6 | 0.097 | 0.076 | −0.440 | $7.49 \times 10^{-4}$ | 0.88 | 78 |
| BM | | | | | | |
| 0 | 0.126 | 0.145 | −0.432 | $3.46 \times 10^{-4}$ | 4.08 | 0 |
| 1 | 0.120 | 0.118 | −0.427 | $2.38 \times 10^{-4}$ | 2.80 | 31 |
| 2 | 0.132 | 0.098 | −0.432 | $2.07 \times 10^{-4}$ | 2.43 | 40 |
| 3 | 0.140 | 0.067 | −0.421 | $1.57 \times 10^{-4}$ | 1.85 | 56 |
| 4 | 0.091 | 0.065 | −0.430 | $1.04 \times 10^{-4}$ | 1.23 | 70 |
| 5 | 0.112 | 0.077 | −0.427 | $1.2 \times 10^{-4}$ | 1.40 | 65 |
| 6 | 0.103 | 0.078 | −0.422 | $1.2 \times 10^{-4}$ | 1.41 | 65 |

### 3.4. Adsorption Isotherms Plots of NP Extracts Adsorption on Carbon Steel Surface

An adsorption isothermal calculation was adjusted to various isothermal equations, namely Frumkin, Langmuir, Temkin, Freundlich, Bockris–Swinkels, and isothermal Flory–Huggins [20]. The variation in NP extract concentration and inhibition efficiency was used as the input parameters. Table 3 and Figure 4 indicate that the best fittings were obtained for the WM and BM zones by using the Langmuir adsorption isothermal equation [21], while the HAZ fits well with the Freundlich adsorption isothermal equation [22].

$$\text{Langmuir equation}: \ \frac{C}{\theta} = \frac{1}{K_{ads}} + C \tag{4}$$

$$\text{Freundlich equation}: \ \log\theta = \log K_{ads} + n\log C \tag{5}$$

$$K_{ads} = \frac{1}{55.5} exp\left(\frac{-\Delta G_{ads}^{\circ}}{RT}\right), \tag{6}$$

where $\theta$, as the surface coverage, is equal to $\frac{IE}{100}$, $K_{ads}$ is the adsorption equilibrium constant [23], and $\Delta G_{ads}^{\circ}$ is the standard free energy of adsorption; there were 55.5 mol of pure water per liter in the solution. Generally, the amount of adsorption free energy up to $-20$ kJ/mol indicates physisorption. The values of $\Delta G_{ads}^{\circ}$ found for this work (Table 3) indicate a physisorption characteristic. A negative value signifies the spontaneous reaction of NP extract adsorption on the weldment surface [23–25].

**Table 3.** Adsorption isothermal parameter.

| Weldment | $K_{ads}$ | $\Delta G_{ads}$ |
|----------|-----------|------------------|
| WM | 45.87 | −19.42 |
| HAZ | 3.51 | −13.06 |
| BM | 0.45 | −25.21 |

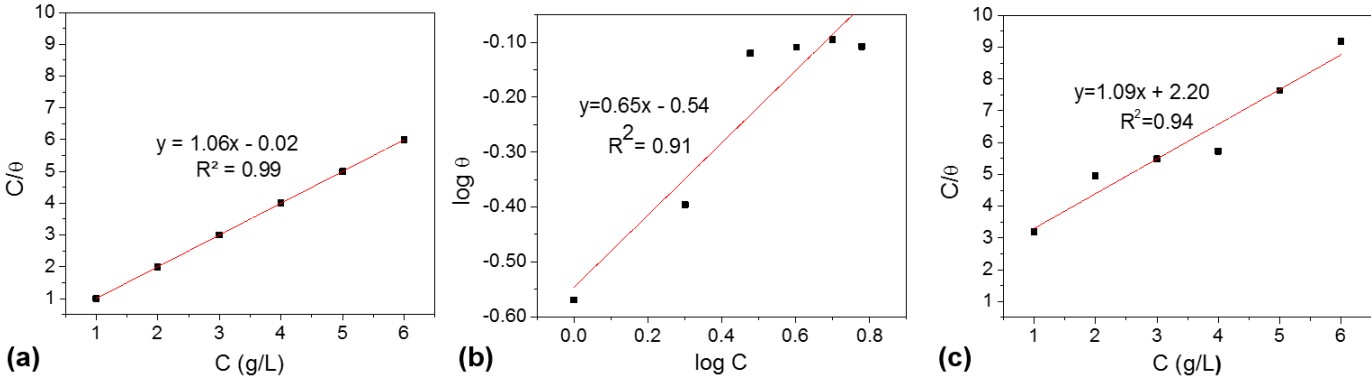

**Figure 4.** (**a**) Langmuir adsorption isothermal curves of WM, (**b**) Frendlich adsorption isothermal of HAZ, (**c**) Langmuir adsorption isothermal of BM.

### 3.5. Electrochemical Impedance Spectroscopy

Figure 5 shows EIS spectra of all specimens tested at different concentrations of NP extract. Based on the proposed equivalent circuit model, corrosion parameters such as polarization resistance ($R_p$), solution resistance ($R_s$), and constant phase element (CPE) were obtained; thus, the inhibition efficiency was calculated as detailed in Table 4. Similar to those obtained from the polarization test, the EIS test results show that the highest inhibition efficiency for the WM zone (76%) was also obtained at an NP extract concentration of 5 g/L. This concentration also gave the highest efficiency of 66% for HAZ, while, for the BM zone, its highest efficiency of 61% was reached at the NP extract concentration of 4 g/L, the same as that obtained from the polarization results.

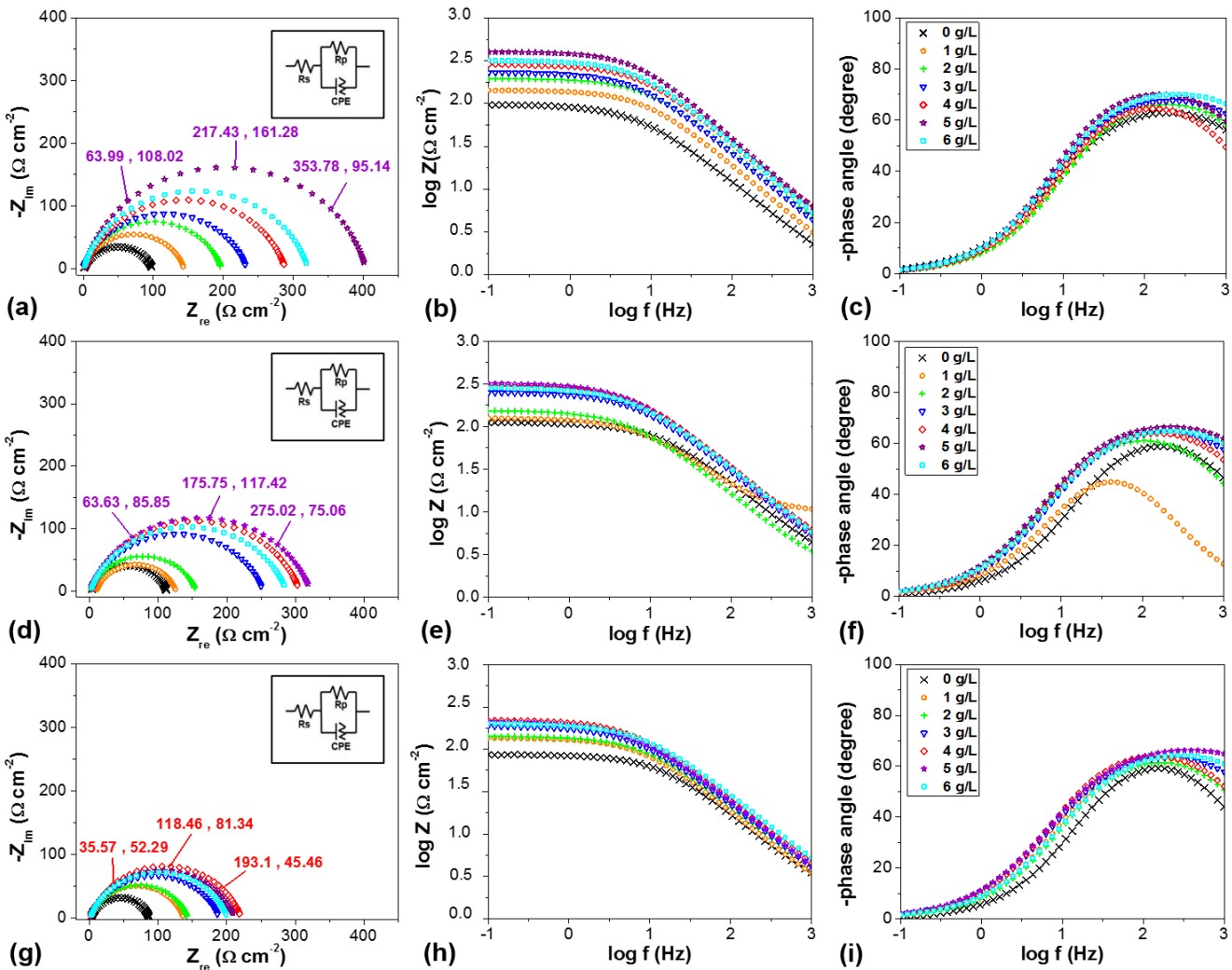

**Figure 5.** EIS spectra of all specimens tested at different concentrations of NP extract, showing plots of Nyquist, Bode modulus, Bode phase angle of: (**a–c**) WM zone, (**d–f**) HAZ zone, (**g–i**) BM zone. A proposed equivalent circuit model is inserted in each Nyquist plot.

**Table 4.** EIS parameters obtained from fitting of the equivalent circuit model.

| NP Extract (g/L) | Rs ($\Omega.cm^{-2}$) | Rp ($\Omega.cm^{-2}$) | CPE ($\mu F.cm^{-2}$) | n | $\chi^2$ | IE (%) |
|---|---|---|---|---|---|---|
| WM | | | | | | |
| 0 | 0.58 | 98 | 491 | 0.789 | 0.285 | 0 |
| 1 | 0.61 | 143 | 243 | 0.829 | 0.327 | 31 |
| 2 | 1.26 | 196 | 169 | 0.83 | 0.156 | 50 |
| 3 | 0.84 | 232 | 183 | 0.824 | 0.099 | 58 |
| 4 | 2.77 | 286 | 134 | 0.834 | 0.169 | 66 |
| 5 | 1.89 | 402 | 97 | 0.862 | 0.035 | 76 |
| 6 | 0.80 | 320 | 128 | 0.839 | 0.122 | 69 |
| HAZ | | | | | | |
| 0 | 1.96 | 110 | 245 | 0.808 | 0.089 | 0 |
| 1 | 1.85 | 129 | 350 | 0.804 | 0.087 | 17 |
| 2 | 1.65 | 153 | 355 | 0.8 | 0.103 | 28 |
| 3 | 1.3 | 251 | 189 | 0.801 | 0.072 | 56 |
| 4 | 2.02 | 304 | 161 | 0.808 | 0.111 | 64 |
| 5 | 0.97 | 321 | 171 | 0.804 | 0.07 | 66 |
| 6 | 1.12 | 286 | 180 | 0.794 | 0.066 | 62 |

**Table 4.** *Cont.*

| NP Extract (g/L) | Rs ($\Omega.cm^{-2}$) | Rp ($\Omega.cm^{-2}$) | CPE ($\mu F.cm^{-2}$) | n | $\chi^2$ | IE (%) |
|---|---|---|---|---|---|---|
| | | | BM | | | |
| 0 | 1.77 | 85 | 285 | 0.825 | 0.148 | 0 |
| 1 | 1.23 | 136 | 279 | 0.815 | 0.222 | 37 |
| 2 | 1.52 | 142 | 260 | 0.803 | 0.128 | 40 |
| 3 | 0.92 | 188 | 273 | 0.786 | 0.066 | 55 |
| 4 | 1.66 | 219 | 212 | 0.814 | 0.054 | 61 |
| 5 | 0.37 | 211 | 257 | 0.785 | 0.047 | 60 |
| 6 | 0.84 | 201 | 206 | 0.789 | 0.036 | 58 |

The increase in NP extract concentration corresponds to the enlargement of the semi-circle diameter of Nyquist plots. It did not, however, influence their shape, indicating that a similar corrosion mechanism was taking place. A charge transfer mechanism appears to control the corrosion process. A constant phase element (CPE) was used to describe the independent phase shift frequency between the applied alternating potential and its current response. The following mathematical expression defined the CPE [26]:

$$Z_{CPE} = \frac{1}{Q_0(j\omega)^n} \tag{7}$$

where Z (CPE) is the impedance of CPE and $\varpi$ is the angular frequency. The numerical value of the admittance (1/ZCPE) at $\omega = 1$ rad/s is owned by $Q_0$, and n is the surface irregularity. Represented by $Q_{CPE}$, a pure capacitance at n = 1 will become a pure resistance at n = 0. The $Z_{CPE}$ value is related to film thickness, surface heterogeneity, and surface roughness; CPE causes a greater depression in the Nyquist semicircle diagram, where the metal– solution interface acts as a capacitor with an irregular surface [27,28].

The semicircles of the WM zone are larger than those of the other zones, indicating a higher surface impedance or $R_p$. The insertion of CPE into the proposed circuit model creates a non-ideal distribution of $R_s$ and double-layer capacity [20]. An increase in the phase angle along with the increase in the NP extract concentration can be related to the decrease in the capacitive behavior of the metal surface due to the dissolution process.

The addition of NP extract up to 6 g/L increased the value of $R_p$. The maximum value of $R_p$ was reached at the concentration of 5 g/L for the WM and HAZ specimens, and at 4 g/L for the BM specimen, indicating an instability in the solution. At the concentration of 5 g/L, the $R_p$ reached its maximum value, while the $R_s$ value did not change, revealing no influence of the NP extract on the solution resistance. The only increase in the $R_p$ value indicates that inhibition only happens on the metal surface. This inhibition prevents the charge transfer from the metal surface into the solution, resulting in a decrease in oxidation and reduction reactions [20]. The increase in inhibition efficiency proves the inhibition process of the NP extract molecules on the metal surface [18,19]. This process leads to the formation of a protective layer on the metal surface, as indicated by the increase in $R_p$ value [20]. As the NP concentration increases, the protective layer thickens at the optimum inhibitor concentration, and eventually breaks off due to a dissolution process [21].

### 3.6. SEM/EDS Observation

Figure 6 shows the surface morphology of the specimens after being immersed for 10 days in 1M HCl and 1M HCl+NP extract. Some cracks and pits appear on the weldment surface after being immersed in HCl-only solution. Meanwhile, a more homogenous corrosion product was formed on all specimens after immersion in HCl+NP extract, indicating that a uniform corrosion was taking place. This observation points out the inhibition effect of the NP extract against corrosion. The EDS analysis revealed an increase in oxygen (Table 5), indicating the formation of some oxides covering the metal surface, probably in the form of $Fe(OH)_2$ [6].

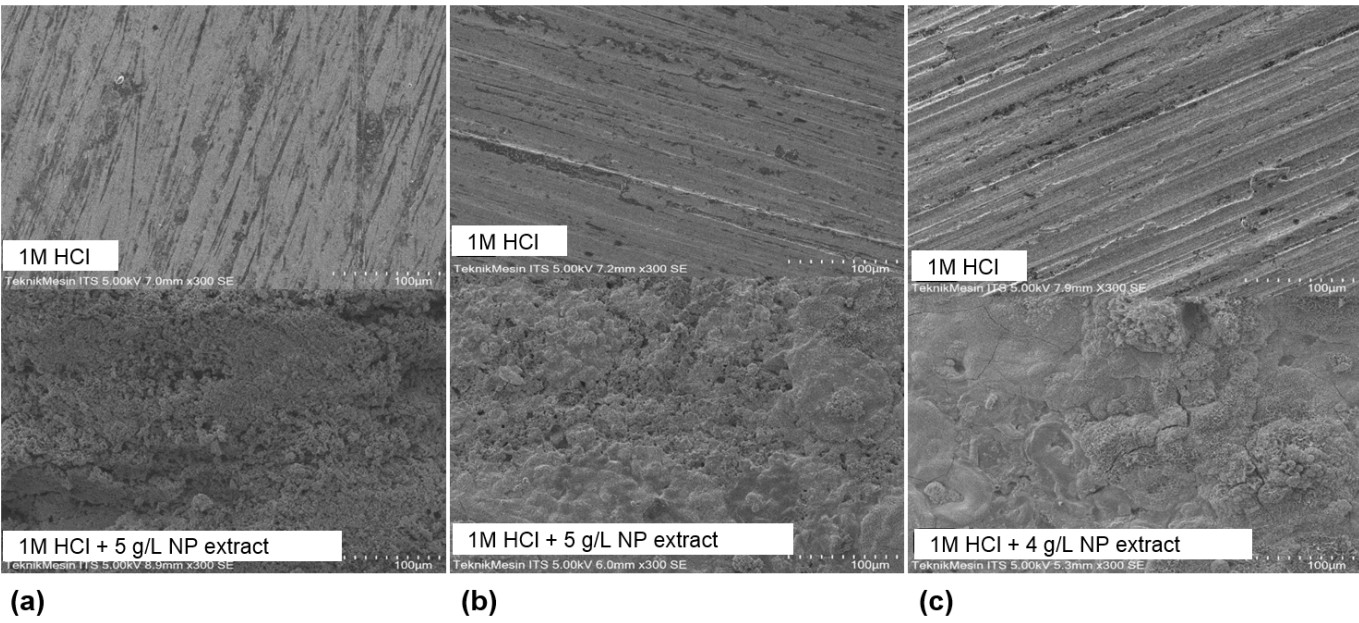

**Figure 6.** Secondary electron mode SEM images of: (**a**) WM, (**b**) HAZ, (**c**) BM, after being immersed for 10 days in 1M HCl and 1M HCl+NP extract (4 g/L for BM, 5 g/L for WM and HAZ).

**Table 5.** Detection of elements on the specimen surface by using EDS.

| Specimen | Element (wt %) | | | | |
|---|---|---|---|---|---|
| | C | O | Na | Cl | Fe |
| WM in HCl | 1.63 | 1.63 | 0.43 | 0.25 | 96.07 |
| WM in HCl+NP extract | 1.84 | 29.04 | 0.21 | 0.19 | 68.72 |
| HAZ in HCl | 1.84 | 1.24 | 0.55 | 0.16 | 96.06 |
| HAZ in HCl+NP extract | 1.91 | 27.87 | 0.16 | 0.34 | 69.71 |
| BM in HCl | 3.82 | 3.65 | 0.71 | 0.36 | 91.45 |
| BM in HCl+NP extract | 2.29 | 26.76 | 0.32 | 0.91 | 69.72 |

Figure 7 shows FTIR and UV-vis spectra of all specimens after being immersed for 10 days in 1M HCl and 1M HCl+NP extract. In comparison to the original NP extra spectra, different bonds appeared in the spectra of weldment specimens (Figure 7a–c). After immersion in HCl, there was a sharp peak at 1628 cm$^{-1}$, indicating a bending vibration of O-H, and weak absorption at 1050 cm$^{-1}$ indicating a bond with Fe (hematite). After immersion in HCl+NP extract, the intensity of these two peaks was reduced, indicating the formation of hematite groups (iron oxide, $Fe_2O_3$) instead of OH bonds (iron hydroxide). In the UV-vis spectra (Figure 7d–f), a shift in the maximum wavelength and a change in absorption intensity were observed in all conditions. The change in the pattern and the shift of maximum λ indicate a reaction or change in chemical composition of the metal surface after the addition of NP extract. The NP extract probably reacted directly with HCl and reduced the acidic content, thus decreasing the dissolved Fe, as marked by the shift in maximum wavelength to the higher λ.

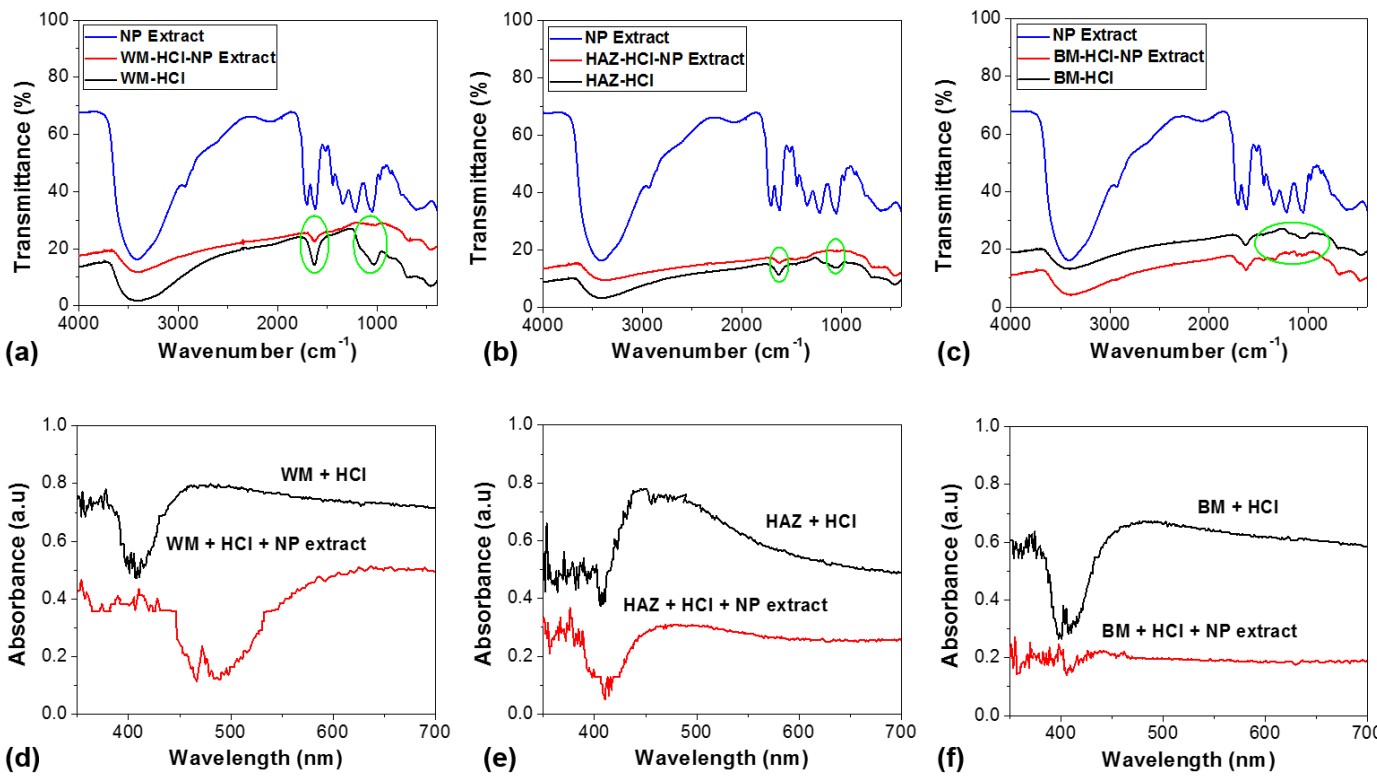

**Figure 7.** (**a–c**) FTIR spectra and (**d–f**) UV-visible spectra of WM, HAZ, and BM zone after being immersed for 10 days in 1M HCl and 1M HCl+NP extract (4 g/L for BM, 5 g/L for WM and HAZ).

## 4. Discussion

The microstructure of the weld metal (WM) zone is dominated by the grain boundary of ferrite (Figure 1), as the result of a low cooling rate and a process of ferrite formation with carbon diffusion. The heat-affected zone (HAZ) is dominated by acicular ferrite and a fine–coarse grain structure. The hardness test results on the three weldment zones reveal that the base metal (BM) zone has the highest hardness. A microstructure with small grain size results in a higher volume of grain boundaries [4], which is likely to be more active in preventing corrosion compared to a structure with a lower volume of grain boundaries, as $Cl^-$ ions are easier to diffuse. A higher volume of grain boundaries raises chemical activities in corroding the surface that promote the formation of a dense passive film with higher stability and protectiveness [4]. In HCl solution, the corrosion rate of WM is the highest because of the largest grain size and the lowest hardness. Low hardness metals are known to have low corrosion resistance [29–31]. In addition, the formed precipitates during the welding process contribute to the highest corrosion rate in the WM zone [32]. These precipitates usually contain non-metallic elements that potentially increase the kinetics of corrosion [32].

### 4.1. Effectiveness of NP Extract as Corrosion Inhibitor

Difference sin grain size makes some areas in the weldment more active than others [3,33]. The addition of 4 g/L NP extract increased the corrosion resistance of BM zone specimens up to 70%. There are changes in the values of βa and βc to positive and negative directions (Table 2), indicating the effect of NP extract on the cathodic and anodic processes. The change in corrosion potential ($E_{corr}$) is not larger than 85 mV in either the positive or negative direction. This implies that, on BM specimens, NP extract behaves as a mixed-type inhibitor [13,18]. Changes in $E_{corr}$ were also found on the HAZ and WM specimens with a less than 85 mV shift; thus, the extract also acted as a mixed-type inhibitor. In all weldment zones, NP extract acts as a mixed-type inhibitor with predominant cathodic effectiveness.

This indicates that the anodic metal dissolution reaction of iron and the cathodic reaction of hydrogen evolution were inhibited after the addition of NP extract to 1 M HCl [34].

The Langmuir (WM and BM) and Freundlich (HAZ) adsorption isothermal calculations indicate the presence of a solid surface which is generally confined to a single molecule layer (monolayer). The Langmuir adsorption is generally applied to a homogeneous surface [21]. Therefore, the heterogeneous HAZ surface has a different adsorption than the other zones. The physisorption reveals an interaction between the molecules and the metal in the form of electrostatic or Van der Waals interactions [35].

The NP extract molecules were supposed to be firstly adsorbed onto the steel surface and blocked the reaction sites of the weldment. In this way, the surface area available for $H^+$ ions was reduced without affecting the main reaction mechanism. A larger NP extract coverage on the WM surface (coarse grained) was reached at higher concentration, and a homogenous corrosion took place as indicated by the anodic polarization curve in the weldment (Figure 3). This demonstrates the effect of NP extract adsorption on the formation of a protective film. This film is usually composed of inhibitor extract molecules, chloride, and iron oxide [34], thereby increasing the iron passivation [36].

The maximum inhibition efficiency of NP extract on the WM zone and HAZ specimens was reached at 5 g/L, while, on the BM specimens, it was reached at 4 g/L. This difference can be related to the different microstructural characteristic of WM and HAZ compared to that of the BM zone. The coarse grain of WM does not promote the formation of a compact protective layer, leading to a more active corrosion condition [4]. However, the adsorption of NP extract compensated this condition by accelerating the formation of a protective oxide layer (Figure 7), as shown by the highest inhibition efficiency. On the BM zone specimen, the intensity of hematite and OH peaks were very low and almost unidentifiable, indicating a weak passivation when compared to what occurred on the HAZ and WM specimens. The BM zone showed a cathodic behavior, having a more noble or less active anodic area, while the HAZ and WM zones behaved anodically with a more active zone of metal dissolution and oxides formation.

### 4.2. Proposed Mechanism of Inhibition

Based on GCMS data (Table 1), mome inositol is the most abundant and dominant compound in NP extract. The hydroxyl group (–OH), having a free electron on the O, could attack empty d orbitals on Fe metal to form a metallic bond. As a result, it was difficult for Fe to release its ions to form $Fe^{2+}$ or $Fe^{3+}$, or, in other words, the corrosion had been successfully inhibited. The iron–mome inositol (NP extract) complex was formed directly based on the donor acceptor interaction between the species' $\pi$ electrons and N atoms, and also the $\pi$ electrons of the cationic species and the vacant iron d orbitals. The complex iron solubility depends on the inhibitor's molecules and the groups of hydroxyls. The solubility also determines the degree of corrosion inhibition. The formed complexes were precipitated at the WM–solution interface and retarded corrosion at optimum NP extract concentration [28,37].

In the presence of NP extract inhibitor, the inhibitor species forms a deposit on the WM surface due to the high electronegative bonds [37]. Initially, this interaction started on the active spots of the WM surface, which could be related to anodic reaction mitigation as shown by the polarization results. As the immersion continued, the interaction spread over the entire WM surface, leading to the formation of a protective film through the following productive adsorption reaction [37].

$$Fe^{2+} + xH_2O + bCl^- + zNPextract \rightarrow [Fe\,(NPExtract)Z(OH)_xCl_b]^{2-x-y} + H^+ \quad (8)$$

This film could have reached its highest compactness and adherence at the NP extract concentration of 5 g/L, as indicated by the presence of $Fe_2O_3$ and FeOOH on the inhibited WM surface [28,37]. The added NP extract molecules were adsorbed on the metal surface and by modifying the less stable iron chloric complexes to the more stable iron-NP extract complexes which replaced the already adsorbed water molecules.

## 5. Conclusions

This work reveals a change in the corrosion behavior of steel weldment after being inhibited by rambutan fruit (*Nephelium lappaceum*, NP) extract. The NP extract increases the corrosion resistance of the weldment in 1M HCl solution with a different efficiency for each weldment zone. The NP extract acts as a mixed-type inhibitor that contributes to the increase of oxide layer stability, thus increasing the corrosion resistance of the weldment. The maximum inhibition efficiency of the NP extract was reached at 97% for weld metal with 5 g/L of extract, at 80% for the heat-affected zone with 5 g/L, and at 70% for base metal with 4 g/L. This work confirms the effectiveness of NP extract as an organic corrosion inhibitor for steel weldment in an acidic environment. Different inhibitor concentrations are needed for different weldment zones to obtain the optimum corrosion protection.

**Author Contributions:** Conceptualization, F.G.; Methodology, F.G.; Formal analysis, F.G., D.B.D. and P.H.S.; Investigation, F.G.; Data Curation, F.G., K.A.M. and H.I.; Writing—Original Draft, F.G.; Writing—Review and Editing, F.G., A.T., H.H.; Visualization, F.G.; Project Administration, F.G.; Funding Acquisition, F.G. All authors have read and agreed to the published version of the manuscript.

**Funding:** This research was funded by "Hibah Penelitian Kolaborasi Internasional (HAPKI)-LPPM UB" grant number: 539.3.4/UB19.C10/PN/2021.

**Institutional Review Board Statement:** Not applicable.

**Informed Consent Statement:** Not applicable.

**Data Availability Statement:** The data presented in this study are available on request from the corresponding author.

**Conflicts of Interest:** The authors declare no conflict of interest.

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
