# Peer review of "Nephelium lappaceum Extract as an Organic Inhibitor to Control the Corrosion of Carbon Steel Weldment in the Acidic Environment"

_sustainability, doi:10.3390/su132112135_

Round 1

Reviewer 1 Report

In my opinion the Manuscript sustainability-1407212 entitled "Nephelium lappaceum extract as organic inhibitor to control the corrosion of carbon steel weldment in acidic environment" is suitable for publication in the journal.

The manuscript includes important issues that may be of interest  to a broad chemical engineering and/or applied chemistry audience. Applied research techniques are adequate to  the studies taken. Language is clear and understandable. Conclusions are related to the obtained test results.

However, there are some remarks and questions related to the presented research results:

  • Line 42 inhibitors form adsoption layer, not passive layer. A passive layer can be formed due to the presence of ihbibitor.
  • Fig 3. It would be better to use lines without points that blur the graphs. The drawings are small and overlap. They are hard to read.
  • In order to use Tafel extarpolation, one must be sure that the graph is linear for at least one current decade. Have the authors checked this relationship?
  • Table 3 HAZ, 1000 mg/L Why such a large, almost fivefold increase in the resistance of the solution?
  • Line 227 The authors suggest the reaction of the inhibitor molecules with HCl. Have FTIR or pH tests been performed on the solution after exposure to an HCl solution?
  • Line 287-293 Why was hematite formation not included in the proposed inhibition mechanism ?
  • 7. The SERS enhanced Raman scattering technique can be used to determine the orientation of the inhibitory molecules relative to the surface. Did the authors consider such studies?
  • It would be advisable to plot an adsorption isotherm for the tested inhibitor.

Reviewer 2 Report

Work accepted with major revision

Reviewer Comment for Author:

  1. In this work there are several mistakes in English, it is necessary to correct and revise them, I give some examples:

- Correct the title by adding:  “an” as an organique and “the” in the acidic

- In the paragraph: 2.1. NP Extract preparation:

A batch of 1500 gram add s “grams”

and then grinded grind is the irregular verb in the past “ground”

the extract substance were replace were by “was”

  1. in your abstract:

 - You say electrochemical and immersion methods, immersion isn’t methods, correct this and add electrochemical methods (Polarization curves and electrochemical impedance diagrams).

- The concentration of your inhibitor varies from 0 to 6000 mg/L, it is better to use these concentrations in g/L (6g/L)

  1. In a paragraph: 2.3. Corrosion testing

- You say, …A three-electrode setting was used with the specimen, an Ag/AgCl, and a carbon rod as the working, reference, and counter electrodes, respectively. Not correct Ag/AgCl a reference electrode, a carbon rod...

- What is the difference between corrosion current density and corrosion current we use icorr not Icorr correct this in your manuscript

- pour les mesures spectroscopiques, nous utilisons une gamme de fréquence varié de 100KHz à 10 mHz pourquoi vous utilisez 1KHZ à 0.1Hz

  1. In a paragraph 3.3. Potentiodynamic polarization

- For this area, the lowest corrosion rate of 0.67 mm/year was reached at the extract concentration of 5000 mg / L, You plot the polarization curves, explain how you calculate the value of 0.67 mm/year and the same for 1.23 mm/year

- Enlarge the polarization curves and correct the ordinate axis log(icorr)

- In table 2, which sign of cathodic slope, correct that

  1. In a paragraph 3.4. Electrochemical impedance spectroscopy

- In the Nyquist representation, make some frequency values appear

- In the title of Figure 4, there are several repetitions

- To calculate the resistances and the capacities, you work always on surface units, for which reason you use the Ω and µF

- CPE is calculated by which relation, added in the text

- The coefficient N represents what, we use n not N

- You say: …an inhomogeneous distribution of Rs and double-layer capacitance ... what did you mean, corrected this and, we say capacity, not capacitance

  1. In a paragraph: 4.1. Effectiveness of NP extract as corrosion inhibitor

- You represent the polarization curves in a shorter interval [-150, 150], it is better to increase it to see for example diffusion phenomenon

- You say "This demonstrates the effect of NP extract adsorption on the formation of anodic protective film. This film is composed of inhibitor extract molecules, chloride, and iron oxide [26], thereby increases the iron passivation". We cannot speak of an anodic film because the inhibitor is mixed and the film formation is made in the stationary state and secondly the existence of chloride ions and iron oxides does not increase the passive film. Justified

  1. How your inhibitor adsorbs onto the metal surface. Added this part
  2. Figure 7 makes no sense, to do this; you need to do the theoretical calculations for the three abundant products in your extract.

Round 2

Reviewer 2 Report

Correct the sign of βc, it is negative

Author Response

On behalf of the Authors I thank you for the opportunity given to us to improve our manuscript. We have revised our manuscript based on the corrections. We have made changes to the manuscript. We believe that the corrections improve the quality of our article. Hopefully, our revision will satisfy the editors and referees.

Femiana Gapsari
